# The State of the Evidence about the Family and Community Nurse: A Systematic Review

**DOI:** 10.3390/ijerph19074382

**Published:** 2022-04-06

**Authors:** Federica Dellafiore, Rosario Caruso, Michela Cossu, Sara Russo, Irene Baroni, Serena Barello, Ida Vangone, Marta Acampora, Gianluca Conte, Arianna Magon, Alessandro Stievano, Cristina Arrigoni

**Affiliations:** 1Section of Hygiene, Experimental and Forensic Medicine, Department of Public Health, University of Pavia, 27100 Pavia, Italy; federica.dellafiore@unipv.it (F.D.); cristina.arrigoni@unipv.it (C.A.); 2Health Professions Research and Development Unit, IRCCS Policlinico San Donato, 20097 Milan, Italy; rosario.caruso@grupposandonato.it (R.C.); irene.baroni93@gmail.com (I.B.); gianluca.conte@grupposandonato.it (G.C.); arianna.magon@grupposandonato.it (A.M.); 3Rsa Attanasio, Rsa Limbiate, 20812 Milan, Italy; michela.cossu01@universitadipavia.it; 4Nursing Degree Course, Section Istituti Clinici di Pavia e Vigevano S.p.A., University of Pavia, 27100 Pavia, Italy; 5Department of Biomedicine and Prevention, University of Rome “Tor Vergata”, 00173 Rome, Italy; ida.vangone01@universitadipavia.it; 6EngageMinds HUB, Consumer, Food and Health Engagement Research Center, Department of Psychology, Università Cattolica del Sacro Cuore, 20100 Milan, Italy; serena.barello@unicatt.it (S.B.); marta.acampora01@icatt.it (M.A.); 7Department of Oncology and Hematology-Oncology, IEO-European Institute of Oncology, 20100 Milan, Italy; 8Centre of Excellence for Nursing Scholarship, OPI Rome, 00173 Rome, Italy; alessandro.stievano@gmail.com

**Keywords:** community nurse, family nurse, patient-centred care, state-of-art systematic review

## Abstract

Introduction. The increase in chronic degenerative diseases poses many challenges to the efficacy and sustainability of healthcare systems, establishing the family and community nurse (FCN) who delivers primary care as a strategic role. FCNs, indeed, can embrace the complexity of the current healthcare demand, sustain the ageing of the population, and focus on illness prevention and health promotion, ensuring a continuous and coordinated integration between hospitals and primary care ser. The literature on FCNs is rich but diverse. This study aimed to critically summarise the literature about the FCN, providing an overall view of the recent evidence. Methods. A state-of-art systematic review was performed on PubMed, CINAHL, and Scopus, employing the Preferred Reporting Items for Systematic Reviews and Meta-Analyses (PRISMA) statement and checklist to guide the search and reporting. Results. Five interpretative themes emerged from the 90 included articles: clinical practice, core competencies, outcomes, Organisational and educational models, and advanced training program. Conclusions. FCNs can make a major contribution to a population’s health, playing a key role in understanding and responding to patients’ needs. Even if the investment in prevention does not guarantee immediate required strategies and foresight on the part of decisionmakers, it is imperative to invest more political, institutional, and economic resources to support and ensure the FCNs’ competencies and their professional autonomy.

## 1. Introduction

Population ageing is a global phenomenon [1], which determines and shapes many societal changes worldwide (at the individual, relational, cultural, and organisational levels), posing novel clinical, psychosocial, and welfare challenges [2]. In Western countries, the elderly population (i.e., older patients, defined as age > 65 years) has swelled due to the increased life expectancy [3]. For example, in Italy, the average life expectancy reached its average age peak of 83.22 years in 2019 [3]. These data represent a great success, but at the same time, a significant societal issue when older citizens cannot survive at optimal levels of their state of health. It is clear, in fact, that after a certain age, people spend about one-third of their life in good health and then develop a chronic disease [4].

Consequently, a further challenge is determined by an increase in chronic conditions in the population, since due to the more rapid spread of diseases, the demand for treatment has increased [5]. Moreover, factors such as stressors, which have led the population to suffer from long-term diseases, have grown considerably [6]. Coping with chronic conditions means dealing with a patient’s long-term management and illness in his or her life context. Chronic patients cannot be hospitalised for a long time to manage their disease; however, they require connection to the health facility through an organisational bridge that fosters communication and exchanges among health professionals and care services [7].

The COVID-19 pandemic has contributed to a greater focus on this societal phenomenon, accelerating the need for an organisational revolution toward service models aimed at supporting the categories of people defined as vulnerable at home [8]. Furthermore, the growing expectations and attitudes of health and assistance service users have led to a greater number of health requests, the demand for a higher quality of services, and the skills necessary to deliver them [9]. Therefore, the health crisis triggered by the COVID-19 pandemic represents the manifestation of the crises of the last decades, which has already highlighted the imbalance between the growth in demand for care and the decline in economic resources [10].

In this context, the World Health Organisation (WHO) underlined the need to implement models centred on primary healthcare (PHC). The WHO defines PHC as an approach which, as close as possible to people’s everyday environment, ensures the highest level of health. PHC focuses on a community’s needs, from health promotion to disease recognition, from prevention to treatment, and, at least, palliative care [11]. For these reasons, a bridge between the community and the health system is vital [11,12]. Therefore, further promotion of home-based care is needed to enhance PHC, reduce the length of stay in acute hospitals and the physical burden of the healthcare professionals that staff them, and meet the demands of older people who prefer to remain in their own homes [13].

In this scenario, the family and community nurse (FCN) is recognised as the key actor for implementing these new models for healthcare delivery, improving PHC and embracing the complexity of the current healthcare demand as it is effectively integrated into a multidisciplinary team [14]. This role is distinct from the ‘community nurse’, who collaborates closely with other health workers and develops plans for promoting healthcare [15]. FCNs are able to address the needs of the family throughout the course of life, paying particular attention to the vulnerable groups in society [15,16]. Indeed, FCNs provide care based on evidence and best practice; they have the knowledge, skills, and competencies to tailor care to individual needs [17,18,19,20], especially for older individuals with chronic diseases. FCNs bolster the transition away from the hospital-centric to the home-centric model for care [21], improving the care continuity of the healthcare system [14,21]. Many authors have demonstrated that FCNs with advanced competencies and autonomy could improve and assure delivery of PHC [22], which is crucial for health promotion and disease prevention through screening services and vaccination programs, the adoption of health-related behaviours (e.g., exercise, dietary modification, smoking cessation), and adherence to medical recommendations for managing chronic diseases [23].

The literature about the FCN is very heterogeneous and diverse, and it has not yet been synthesised; accordingly, a critical summary of the state of the evidence about the FCN represents a current gap in the literature. The lack of an overall view of the evidence available about FCNs undermines identifying aspects that need further improvement or progression, especially in the organisation and implementation of FCN models. The nursing care management to the family and the community could draw support from a general point of view of evidence that highlights an analysis of its current strengths and weaknesses. In addition, the progress of research may falter if researchers fail to recognise the areas that require more empirical investigations to address gaps in knowledge related to the FCNs. Accordingly, this systematic review aimed to critically summarise the state of the evidence of the FCN, providing a big-picture view of the recent evidence.

## 2. Methods

### 2.1. Study Design

This systematic review was conducted following the criteria, design, and methods of the state-of-the-art reviews [24,25] to provide an overall view of recent evidence on FCNs. Specifically, state-of-the-art reviews focus on describing what is currently known for a given topic [22], examining a broad and heterogeneous body of available evidence on a specific topic, and using the rigorous methodology of systematic reviews [24,26]. Systematic reviews adopt a replicable, scientific, and transparent process that minimises bias through exhaustive literature searches of published and unpublished articles and provides an audit trail of the reviewers’ decisions, procedures, and conclusions [27].

For this study, two independent researchers (MC and SR) systematically sought, appraised, and synthesised research literature evidence between November 2021 and December 2021, adhering to guidelines on the conduct of a systematic review [24,25] and to the ‘Preferred Reporting Items for Systematic Reviews and Meta-Analyses’ (PRISMA) statement and PRISMA flowchart (Figure 1) [28]. The diagram depicts the flow of information through the different phases of a systematic review, mapping out the number of records identified, included and excluded, and the reasons for exclusions [28]. The PRISMA 2020 statement provides a checklist to evaluate the introduction, method, results, and discussion section of a systematic review report, ensuring its rigour.

However, the considerable heterogeneities of methods and aims of the included articles did not allow the researchers to perform a meta-analysis. The researchers used a three-stage approach to the interpretative synthesis of available evidence about FCNs, their competencies, and their practical knowledge. First, MC and SR performed free line-by-line coding of the results from the data extraction of the primary articles; then, the authors grouped these codes into representative units of their meanings. This allowed the generation of five descriptive themes. Finally, descriptive themes were discussed among the entire group of authors using an interactive approach and to critically interpret their meanings, generating our themes [29]. 

### 2.2. Literature Search Strategy

The systematic literature search strategy was conducted independently by two researchers (MC and SR) and performed in different phases, and a third researcher (FD) was involved in resolving any doubts or discussions. First, the databases PubMed/Medline, Scopus, and CINAHL were searched using ‘background queries’ to identify the broad literature, including the keywords ‘Community Nurse’ or ‘Family Nurse’ in the title or abstract. In addition, an open search was conducted on Google Scholar to ensure the inclusion of all available literature. After that, specific foreground queries were structured with keywords, Medical Subject Headings (MeSH) terms, and Boolean Operators, and guided by the following questions: ‘What is the current knowledge about FCNs?’; ‘What are FCNs’ specific skills and competencies?’; ‘In which countries and clinical areas are FCNs most developed?’; ‘What is the impact of FCNs’ competencies on patients, families, and communities’ health outcomes?’.

Accordingly, the concepts arising from the background search were operationalised into a framework-based search strategy based on the SPIDER approach for systematic searches (Sample, Phenomenon of Interest, Design, Evaluation, Research type), where: Sample = nurses; Phenomenon of Interest = family and community nursing (FCNs); Design = every study design; Evaluation = experiences, patient-related, nurse-related, and system-related outcomes; Research type = quantitative, qualitative, mixed-method, and ‘other’, such as commentaries. In our study, SPIDER was used instead of PICO (Population/problem, Intervention/exposure, Comparison, and Outcome) because, including different methodologies, PICO was not applicable [30]. 

The inclusion criteria for retrieving articles were: (a) a focus on FCNs (i.e., a nurse with specific competencies to address the needs of the family throughout the course of life and with particular attention to the vulnerable groups in society); (b) published from the year 2000 onward; (c) published in English, Italian, or Spanish; and (d) availability of the full text and abstract. Articles that focused on FCNs during the COVID-19 outbreak in paediatric clinical settings or with a low-quality appraisal were excluded. Specifically, the authors decided to exclude articles focused on paediatric FCNs because such nurses have different and more specific competencies, and articles on FCNs during the COVID-19 outbreak described a particular period that could introduce bias into our state-of-the-art revision. Finally, to identify additional articles, backward and forward citation tracking was also carried out by examining the reference lists (citation chasing) of included articles. 

Figure 1 shows the flowchart of search strategy: 3098 records were retrieved in the identification phase, and 2858 were removed after screening for duplicates. Accordingly, 235 articles were screened by evaluating the titles and abstracts, 127 articles were removed because their content was not focused on FCNs, and 11 were not available. At the end of the screening phase, three articles were excluded for low-quality appraisals, and three of them were irrelevant for the topic. Then, 97 articles were retrieved in full text and assessed using the critical appraisal checklist ‘Joanna Briggs Institute Qualitative Assessment and Review Instrument’ (JBI-QARI) [31] and a critical appraisal tool [31,32]. After the quality appraisal, 12 articles were excluded as lower quality (*n* = 3) or because their overall content was not focused on FCNs (*n* = 9). Therefore, in the third phase (i.e., inclusion), 90 articles were included in our systematic review, 85 from the databases and 5 from other methods.

### 2.3. Quality Appraisal

The quality appraisal of the 97 eligible articles was performed adopting the JBI-QARI, acknowledging the high heterogeneity of methods [31]. The JBI-QARI appraisal allows the evaluation of the methodological quality of eligible articles having different methodologies. It assesses full-text articles, determining which should be excluded due to low quality. Specifically, the JBI-QARI assesses the risk of bias in the quantitative published research, which helps to emphasise the rigour of the research and level of transferability for qualitative evidence [32].

Two authors independently conducted the quality appraisal process (MC and SR), and the final appraisal grading was based on the overall score computed by summing only the items with a positive assessment. The grading was reported as high, medium, or low quality [32]. Any disagreements between the reviewers were resolved by consensus or referred to a third reviewer (FD). This phase excluded 12 articles as lower quality (*n* = 3) or because their overall content was not focused on FCNs (*n* = 9). The remaining 90 articles (85 from the databases and 5 from other methods) showed moderate or good quality [32], and therefore were included (Figure 1).

### 2.4. Data Abstraction, Analysis, and Synthesis

Appendix A synthesises the main characteristics of the 90 articles included by the previous review phases, showing first author and publication year, title, aim, study design, population, country, theme, and results. Then, according to Greenhalgh and colleagues’ recommendations, the articles’ text and results were descriptively synthesised using the thematic aggregation of the line-by-line coding, initially developing a meta-narrative descriptive theme [33].

Finally, the authors discussed the descriptive themes for interpreting the current knowledge and state of evidence about FCNs, including priorities for future investigation and research. At the end of this process, five descriptive themes emerged: (a) clinical practice, (b) core competencies, (c) outcomes, (d) organisational and educational models, and (e) advanced training program.

## 3. Results

The state-of-the-art systematic review allowed the identification of 90 articles that, in line with the inclusion and exclusion criteria previously identified, focused on FCNs. Specifically, most of them were conducted in European countries, including 40 performed in the UK and 8 in Italy; other contexts included Australia (N = 6), North and South America (N = 6), and Asia (N = 6). We identified nine articles using a semi-experimental or experimental design. Eight adopted quantitative observational cross-sectional data, while 12 articles were conducted with a qualitative design and 5 with mixed-method designs. Finally, a project highlighted FCN formation [34], and numerous discussion papers [34,35,36,37], case reports [17,38,39], and case studies were included [40,41,42,43,44,45,46,47]. 

According to a meta-narrative approach [33], the results of the 90 included articles were analysed and synthesised into five main interpretative themes: (a) clinical practice, (b) core competencies, (c) outcomes, (d) organisational and educational models, and (e) advanced training program. The articles’ results created more than one theme. Even if there was a dominant theme emerging from each article, the results of many articles fell under multiple themes (Table 1). Appendix A shows the main aspects and synthesis of data extraction for each included article.


**(a)** 
**Clinical practice**



The theme ‘clinical practice’ emerged from 54 articles included in our state-of-the-art systematic review, describing the main clinical settings where FCN numbers have grown. The recent literature has shown that FCNs utilise their advanced skills and competencies in health promotion, focusing on primary, secondary, and tertiary prevention. To ensure a higher level of professional intervention in healthcare promotion, FCNs must collaborate and become a strategic bridge for linking the competencies of different health care providers, guiding them to answer the healthcare needs of patients, families and communities [48,49,50,51].

According to the WHO, PHC is a whole-of-society approach that addresses the majority of a person’s health needs throughout their lifetime, focusing on people rather than disease [52]. The literature shows that PHC is the main area where FCNs feel better represented in their key role. FCNs are mainly engaged in promoting oral care and stopping smoking [35,44,49,53,54], providing education for correct blood pressure measurements, and preventing cerebrovascular events [46,55]. Furthermore, the FCN’s engagement is directed to correct oral intake, especially in the elderly, to prevent diseases related to malnutrition and chronic diseases [14,53,54], such as cancer [56]; to reduce antibiotic resistance [45] through the FCN as prescriber [39]; and to reduce elderly abuse [57]. In addition, in the USA, FCNs are recognised as fundamental to advising women in breast augmentation, supporting them during the path through surgery, and providing appropriate counselling [58]. However, the PHC needs of the individuals, families, and communities are different from one country to another. For example, in Africa, patients and communities must be educated about tuberculosis infection [59], unlike in developed countries, which have eradicated the virus. In this regard, Roden and colleagues illustrated how geographical context has an impact on the FCN for promoting health, where the holistic view is different from a rural to an urban context [60]. Still, educational interventions through their clinical practice are the ones that better determine changes in behaviour [61].

FCNs have an important role in patients’ education to support secondary healthcare, especially for chronic diseases. FCNs provide care with advanced autonomy, responsibility, and expertise in the whole social setting, especially for supporting the timely recognition of chronic diseases and preventing their spread [14,62,63]. The literature affirms that the advanced competencies of FCNs equip them to better understand patients’ needs and know them holistically. For example, different articles have demonstrated a high-level control of FCNs over chronic diseases in communities in Israel [64].

In addition, FCNs provide tertiary health care, supporting patients, families, and communities in illness management in different clinical settings, such as vascular disease management [65], psoriasis and eczema [18,66], fibromyalgia [67], arthritis [68,69], and chronic pain [70,71]. Review results have demonstrated how FCNs face challenges in the entire community in tertiary health care, especially for particular and fragile categories [72,73,74]. Chronic diseases, especially diabetes mellitus and chronic heart failure, require a high level of clinical practice and competencies of FCNs [37,75,76] to care for complications of these illnesses in the elderly and in oncology patients [40,77,78,79,80,81,82,83]. In psychological and psychiatric diseases, FCNs ensure patient safety and protect their independence [43]. They ensure patients’ respect and dignity; therefore, communication skills have emerged to be a key characteristic of FCNs [84,85], covering a key role in palliative care [85,86,87,88,89], especially for supporting patients and their families in the end-of-life process.


**(b)** 
**Core competencies**



The core competencies of FCNs are an important topic. At present, a heated debate surrounding FCN core competencies is active in the scientific community, accounting for 20 articles in the state-of-the-art review. According to the integrated model of nursing competence proposed by Caruso and colleagues (2016), nurse competence is an integrated model based on individual characteristics, composed of motivation, education, and the function or tasks that nurses have in the working organisation and focused on nursing-sensitive outcomes (NSOs), which influence nurses’ performances and clinical patients’ outcomes [90]. The literature review depicts FCNs’ core competencies, especially based on health education and health promotion, high communication and empathy skills, and advanced clinical knowledge [38,49].

Many authors have agreed on health education as the FCN competency to enhance and promote family and community health [17,44,49,91,92,93,94,95,96]. It is very important to involve the caregiver in the health education process [97]. FCNs must orient their activities toward self-care to achieve the greatest possible autonomy in the health of patients, improving their attitude and knowledge, and helping them develop the skills needed to achieve better control of themselves [97]. The success of health education depends on FCNs competencies in high communication and empathy skills [45,58,66,96,98,99]. FCNs must talk and listen to their patients without judgment, working hard to deliver impartial and individualised care with the utmost compassion and sensitivity [97,100]. FCNs need advanced clinical knowledge to manage, monitor, and evaluate health activities [95,96]; implement strategies that promote continuity and quality of care; evaluate the impact of care by defining process indicators and results; and have sufficient human and materials resources [101]. In addition, FCNs demonstrate competencies to collaborate and work in multidisciplinary teams. FCNS take the initiative and employ systems thinking to ensure the quality of nursing care [14,92,94,99], and achieve better outcomes by providing culturally appropriate and safe care [95,96,102]. Finally, FCNs must be well prepared and trained to provide care based on evidence and best practice [17,92].


**(c)** 
**Outcomes**



Twenty-one articles, four of which were randomised controlled studies [20,61,74,76,103], emphasised the positive influence of FCNs on patient-, nurses-, and health system-related outcomes. Regarding patient-related outcomes, we found that the intervention or care delivered by FCNs significantly increased survival [42,76,103], improved quality of life and clinical outcomes [38,104,105], and reduced the re-hospitalisation rate [42,103]. Furthermore, Ippoliti and Falavigna demonstrated, through socioeconomic planning, how the FCN’s intervention can determine positive patient-related outcomes, such as those involving hip fractures [101,106]. In particular, educational intervention by FCNs allowed a better adherence to correctly measuring blood pressure to prevent cerebrovascular diseases [55]. Moreover, patients reported being very satisfied with the care provided by FCNs [61], as they felt their psychological and social needs were considered [20]. In fact, mental health is one major aspect of sustaining patients’ well-being, particularly in cancer patients [107], especially to reduce anxiety and stress [74].

FCNs have a new vision in the nursing profession, allowing positive nurse-related outcomes. In fact, FCNs reported a high level of work satisfaction, which is possible because of the continuity of extended holistic care [83,108,109], and patients’ and their families’ full involvement in the educational programs [82,85,110]. However, the quality of education provided by FCNs is related to the geographical context; for example, FCNs feel poorly educated and trained in mountain communities [60]. A higher patient education level by an FCNs’ training program was protective in terms of reducing the risk of using health services and consequently improving system-related outcomes [61,75]. Furthermore, the ‘teach-back’ method, consisting of educational interventions at home, allows reducing public health system interventions and costs [61,75]. The geographical context could impact the delivery of health promotion, specifically for the elderly, who have more difficulty using health services [101].


**(d)** 
**Organisational and educational models**



The literature describes different organizational and educational FCNs models, illustrating how caring patients are placed at the centre of all implemented models. The hospital-at-home model was implemented in elderly care, allowing multidisciplinary and holistic care for patients at their homes [41]. The HADPIPE model was developed to better assist patients, based on a holistic nursing assessment involving psychological, social, and physical aspects [111]. The social aspect is mainly investigated in the family nursing model, which aims to involve the families and caregivers of patients for optimal care delivery [56]. A new FCN model, derived from the Dutch Buurtzorg model, was created to better sustain patients in their context, aiming to develop relationships between FCNs and local care services, and promote independence among patients [112]. Morin and colleagues developed the care management model for patients’ cardiac surgery post-intervention, following patients even after discharge from hospital in highly specialized areas [109]. In Italy, the CoNSENSo project developed model care, illuminating the necessary conditions for the continued provision of health services, and the crucial role of the FCNs to better support the daily activities of elders and their families [101,106,113]. To better understand the importance of FCNs sustaining patients and their families in the educational setting, the FN-AIM model created a map to help nursing students assess patients’ and their families’ knowledge of best practices for maintaining patient health or managing their specific conditions [113,114].


**(e)** 
**Advanced training program**



An FCN is a clinical nurse specialist with a high ability to introduce depth and innovation in care and nursing through evidence-based clinical specialist knowledge, skills, and competencies, requiring specific and advanced training [34,84,115,116]. The articles in our analysis showed the importance of such training programs for FCNs, which are heterogeneous and different from one country to another. In Africa, for example, the nursing council and nursing faculty assured courses for the graduate level, and provided FCNs with the didactic and clinical knowledge, skills, and abilities required for successful FCN practice [117]. Their courses focused on expected learning outcomes, course content, teaching and learning activities, and assessment measures. A graduate course lasts 1 year in China, and allows registered nurses to specialise in family and community nursing [118]. In the USA and Spain, an FCN training program has been proposed during the bachelor’s degree [93,114], providing possibilities to broaden the students’ perspectives to different social groups [119]. In the UK, the development of the FCN competencies allows the nurse to become a prescriber through a 10-day training program following 6 months of experience in the field [39].

Only one article was found describing the training program of FCNs in Italy: a first-level master in a postgraduate course in ‘Family and Community Nursing’ allows nurses to also specialize in telemedicine, which is developing year after year [38]. The training program also trains FCNs in end-of-life care [38]. This could help preserve patients’ dignity, whose significance cannot be underestimated, and could enable FCNs to better care for patients and their families at home [47].

## 4. Discussion

To the best of our knowledge, this systematic review is the first to provide a synthesis of the current state of the evidence about FCNs. According to a meta-narrative approach [33], the results of 90 included articles were analysed and synthesised into five main interpretative themes: (a) clinical practice, (b) core competencies, (c) outcomes, (d) organisational and educational models, and (e) advanced training program. These new themes provide an overview of the synthesis on FCNs. Our review identifies current strengths and weaknesses that need further improvement, especially in the organisation or implementation of FCN models, which slows the progression of research.

FCNs are a valuable and strategic resource in many clinical and care settings. The implementation of FCNs continues to develop day by day, with diverse development concerning the context and healthcare needs of the population [19,64]. Accordingly, our state-of-the-art systematic review included 90 articles from which five interpretative themes emerged, underlining that FCNs can contribute to population health, play a key role in understanding and responding to patients’ needs, and positively influence patient-, nurse- and healthcare system-related outcomes. Finally, our results allowed us to draw important conclusions about where to concentrate future resources in the form of economic investments and organisational improvements. Above all, our results indicate necessary areas for future research, such as the organisation or implementation of FCN models.

Increasingly rich and irrefutable evidence recognises that promoting healthy lifestyles improves health outcomes even in the case of diagnosed diseases and palliative care, especially reducing recurrences of cancers [120,121]. Therefore, PHC—acknowledged as the cornerstone of a sustainable health system by the WHO [11]—presents a golden opportunity for deploying FCNs to apply their advanced competencies and health promotion activities. Considering that scaling PHC interventions across low- and middle-income countries could save 60 million lives and increase the average life expectancy by 3.7 years by 2030 [11], it is clear that FCNs could have a significant impact on the processes of improvement if economic, organisational, and political supports were implemented to allow FCNs to delivery care and utilise their competencies [122]. Moreover, our results show the strategic influence of FCNs’ activities on secondary and tertiary health prevention, further strengthening their strategic role in attaining efficient and sustainable healthcare systems. However, FCNs seem to not yet feel fully at ease with this new role and perspective, demonstrating greater mastery and consolidation in the hospital setting and in disease care [96].

In order to achieve the abovementioned objectives from the salutogenic perspective, FCNs must reach advanced competencies through advanced specialised training [123]. Moreover, organisational policies must accompany FCNs’ competencies and training to allow the birth and implementation of FCN models to provide quality care, responding to the peculiarities of context and the population’s health needs. Indeed, ‘core competencies’, ‘organisational and educational models’, and ‘advanced training program’ are three themes that emerged from our review. Even if they underline a very heterogeneous result [50,123], they comprise a very powerful triad whose optimal balance seems to benefit health outcomes tremendously. The last theme that emerged from our review describes the positive influence of FCNs on patient-, nurses-, and health system-related outcomes. The empirical evidence on this topic is growing, but it is currently fragmented, highlighting an area where further efforts must be made.

According to Longhini and colleagues [122], possible NSOs to the family and community nursing care, and the consequent creation of the Nursing Minimum Data Set (NMDS), have not yet been described. NSOs can be defined as changes in an individual’s state, behaviour, or perception that are measured in response to nursing interventions [124]. The information must be easily registered and systematised in datasets that present specific characteristics to highlight nurses’ contributions to NSOs, in particular, in clinical fields such as family and community settings. For example, the NMDS was created as a minimum set of information that presents standard definitions related to nursing that enable the analysis and comparison of nursing data across different populations, settings, geographic areas, and time [125]. NMDSs have been developed in many countries to determine resource allocation, stimulate research, and guide health policy decision-making [126]. Moreover, complex clinical fields such as family and community care—due to their intrinsic characteristics and burdens—could benefit from a system synthesising the tremendous amount of data from the NSOs in a structured dataset. Data-mining algorithms have the potential to discover meaningful pieces of information and tendencies in this vast net of data that could be transferred into direct knowledge in the clinical field [127].

### 4.1. Strengths, Limits, and Further Challenges

The research design used is the main strength of our study, as it allowed an overall of view on FCNs, depicting the state of evidence that could serve as a booster for designing in-depth future research [24]. However, the high methodological heterogeneity of the included articles, which were mainly performed in the UK, represents our study’s principal limit. Indeed, the practice of FCNs varies so significantly between countries that is it is hard to provide a unified, global definition of what an FCN is. This difficulty necessitates additional studies on FCNs in specific countries, solving the problem of high heterogeneity and providing much more detailed and in-depth information. Moreover, the search terms, including “community nurse” and “family nurse”, are not synonymous across countries/regions, and it may lead to a mix of advanced nurses and generalists having different competencies and different practice scopes. Therefore, a meta-analysis for testing the cumulative effects of implementing FCNs on specific outcomes is difficult to realise, and generalizing the results to other contexts requires caution.

### 4.2. Implications for Clinical Practice

As stated above, our review’s chosen design has allowed us to identify areas where it is necessary to concentrate future resources and experimental research to determine the FCN’s effectiveness on multilevel outcomes. The use of FCNs is progressing in patches; the study of the characteristics of a functional context for FCNs’ implementation could guide policies and organisations in areas where FCNs are not present. Furthermore, assessing the economic savings due to the FCNs’ competencies and skills after their implementation in a specific healthcare system would be very useful. Finally, the nurses’ digital skills are essential for future professional development; however, no study in our FCN review has investigated this aspect.

## 5. Conclusions

The FCN was introduced in 1998 by the WHO, aiming to extend the care of nurses from the individual to the family and community, and from hospital care to primary care services within each community. Many worldwide healthcare systems have undergone changes to implement and support the birth of this new paradigm of nursing care centred on health promotion and prevention. Increasing empirical evidence demonstrates that FCNs, delivering primary care, could embrace the complexity of the current healthcare demand, sustain the ageing population, and focus on illness prevention and health promotion, ensuring a continuous and coordinated integration between hospitals and primary care services. Thus, FCNs can contribute to population health, play a key role in understanding and responding to patients’ needs, and positively influence patient-, nurse-, and healthcare system-related outcomes. Even if investing in prevention does not guarantee immediate required strategies and foresight on the part of decisionmakers, it is imperative that major political, institutional, and economic investments further support and ensure FCNs’ competencies and their professional autonomy to ensure their efficacy and the sustainability of healthcare systems.

## Figures and Tables

**Figure 1 ijerph-19-04382-f001:**
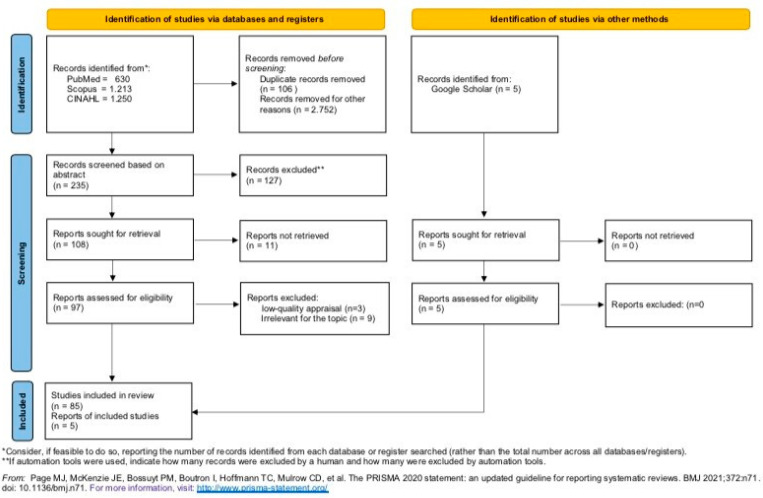
PRISMA 2020 flow diagram for new systematic reviews, including searches of databases, registers, and other sources.

**Table 1 ijerph-19-04382-t001:** Aggregation between included studies and interpretative themes arising from the analysis.

First Author	Year	Clinical Practice	Core Competencies	Outcomes	Organizational and Educational Models	Advanced Training Program
Adamson E.	2013	X				
Adderley J.	2015	X				
Aldridge A.	2014	X	X			
Andrews N.	2021	X				
Bagnasco A.	2020		X			
Balestra M.	2019	X				
Bidone S.	2021		X	X		X
Bright T.	2019	X	X			
Broekema S.	2020			X		
Cavada L.	2021				X	
Chamberlain V.	2016	X				
Chamorro A.	2014		X			
Chater A.	2019	X	X			
Clare C.S.	2017	X				
Connolly M.	2018					X
Courtenay M.	2018	X				X
Cramm J.M.	2017			X		
Daly B.	2015	X				
Davis W.A.	2013	X		X		
Dening H.K.	2016	X				
Dlamini C.P.	2020					X
Duncan D.	2021	X				
Falavigna G.	2020			X	X	
Flavell T.	2015	X				
Fu W.	2010					X
Gafas González C.	2017	X	X			
Green S.M.	2014	X				
Gregg S.R.	2019	X				X
Holdoway A.	2019	X				
Howerton C.R.	2011	X	X			
Husband J.	2008	X				
International Family Nursing Association (INFA)	2020	X	X			
Ippoliti R.	2018			X	X	
Jin L.	2020	X		X		
Johnson A.	2015	X				X
Johnson A.	2015	X		X		
Jongudomkarn D.	2014	X		X	X	
Kelly A.M.	2019	X		X		
Kent S.	2011	X				
Kwok T.	2008	X		X		
Lalani M.	2019				X	
Lee G.	2017				X	
Looman W.S.	2020				X	X
Lumbers M.	2021	X				
Ma W.	2018	X				
MacDonald J.M.	2005				X	
Marcadelli S.	2019	X	X			
Martinez-Riera J.R.	2019	X				
McCrae N.	2014	X				
McKenzie J.E.	2007			X		
Mendes A.	2018		X			
Mendes A.	2017	X				
Mnisi S.D.	2012	X				
Morin D.	2009				X	
Nazarko L.	2013	X				
Nazarko L.	2016	X				
Nissanholtz-Gannot R.	2020	X				
Nissanholtz-Gannot R.	2017			X		
Norman K.M.	2015		X			
Ogston-Tuck S.A.	2018	X				
Oliver S.	2009	X				
Omeri A.	2004		X			
Østergaard B.	2021			X		
Papadopoulou C.	2021					X
Papathanasiou I.	2020					X
Phelan A.	2010	X				
Phillips A.	2016	X	X			
Pickstock S.	2017	X				
Pisano González M.M.	2019		X			X
Randall D.	2021					X
Reid J.	2014	X				
Roden J.	2016	X		X		
Sasso L.	2018		X			
Savini S.	2021	X		X		
Simonetti V.	2021	X		X		
Skingley A.	2016	X				
Slevin E.	2003	X				
Stuart E.	2020	X				
Terracciano E.	2020			X		
Huy N.V.	2018		X			
Vogel R.G.	2021			X		
Wacharasin C.	2008		X			
Wang S.	2019	X				X
Weber S.	2010	X				
Widyarani D.	2020		X			
Wilkes L.	2014		X			
Wood-Baker R.	2012			X		
Yates A.	2019	X				
Yingling C.T.	2017					X
Zimansky M.	2020			X		
Total number of studies	90	54	20	21	9	14

## Data Availability

Not applicable for the methodology of the study.

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
