# Peer review of "The State of the Evidence about the Family and Community Nurse: A Systematic Review"

_ijerph, 2022, doi:10.3390/ijerph19074382_

Round 1
Reviewer 1 Report
Thank you for the opportunity to review this interesting paper about Family and Community Nurse current evidence. There are some appreciations that would improve the manuscript:
Title
According to PRISMA Statement, I suggest adding “systematic review” to the title.
Abstract
Line 22 and follows - Please, consider rephrasing of sentence (it is too long and difficult to read).
Line 30 – How many articles were finally included in this review? It is stated 91, but the flowchart and other sections of the manuscript refer 90. This needs clarification.
Introduction
The introduction is based on very recent and relevant literature. It is rather log but provides a good rational for the study.
Please, use the same font in whole manuscript. I am not sure if italics in lines 76-89 are appropriate used; thus, I would recommend remove it.
Methods
In general, the authors provide detailed information about how the study was conducted.
There are only two minor comments:
Please, could the authors provide more detailed information about state-of-art design?
The time when the search was performed is missing.
Results
Please, check table 1 percentages (the total sum is greater than 100%). Also, I highly recommend detail authors, year, design, sample & instrument and outcomes in another table. This would provide relevant information to readers about the studies included and definitively, it would encourage the quality of this paper.
Discussion
The authors discuss their findings in the context of the current literature.
It is common to state the aim of the study in the first paragraph of the discussion.
Line 364-374 This sentence could be deleted
Implications for clinical practice should be stated in a final paragraph.
Conclusion
I am not sure about suitability of citation in text in this section. It would be more convenient to focus this section on the main authors’ conclusions that respond to the proposed study objectives.
References
Please, check reference list. Some of the entries has no year of publication in bold.
Author Response
Dear Editor and reviewers,
Thank you for the opportunity to review our manuscript for consideration in the IJERPH. We took into account all the points suggested in order to improve the quality of the paper. Please find below point-by-point answers to the recommendations. Changes are highlighted in red in the track version of the manuscript.

Reviewer 2 Report
This was a systematic review of the evidence related to family and community nurse. The authors conducted comprehensive search and synthesized the findings. Although the content was good, it was unclear the difference between 'community nurse' and 'family and community nurse'. The authors MUST address the difference in the introduction to highlight the importance of this manuscript.
Introduction:
Line 58-89, please summarize the content and describe what was 'family and community nurse' and the difference to 'community nurse'. You may read a related definition from the American Nurses Association, https://www.nursingworld.org/practice-policy/workforce/public-health-nursing/
Line 90, please give example to support the claim that "The literature about the FCN is very heterogeneous and diversified". The WHO document, Enhancing the role of community health nursing for universal health coverage (2017), has stated the core competencies required in community health nursing.
Methods:
Line 140, FNC, a typo error? Please check the whole manuscript.
Line 145, I suggested the phenomenon of interest as 'family and community nursing'
Line 151, 'focus on FCN', the authors should give a clear definition of FCN for this manuscript.
Line 152, please explain the reasons that studies before 2000 were not included.
Line 172-187, the JBI-QARI was used to assess the quality of the potential eligible studies with different methodologies. Line 177 stated that JBI-QARI might have some items for quantitative research only. Would the items affect the overall quality of qualitative research? Please state how many items were in the JBI-QARI and how to categorize as high, medium, or low quality.
Line 184, the authors indicated that 11 articles were excluded after quality appraisal. However, only three articles were indicated as 'low-quality appraisal' in Figure 1. Please check the accuracy.
Results:
Regarding the definition of primary health care, what was the difference between Line 228-230 and Line 68-71? Both were from the WHO.
Please check the correctness of number in Table 1 and the content. For example, there were 53 articles related to clinical practice in Table 1. But the author stated there were 57 articles in Line 220.
Please provide a brief definition of each emerged theme.
Discussion:
Line 364-366, please use the original article from WHO.
Please add a paragraph to summarize the findings.
Line 384-386, please give example what areas needed to concentrate in future, and how to address the needs?
It was unclear if the findings can answer the proposed question in Line138-141 appropriately.
Author Response

(The authors gave the same response as above.)

Reviewer 3 Report
I appreciate this opportunity to review this article. I have some major concerns and some minor concerns with this manuscript. Upon multiple rereads, I would suggest limiting the scope of this review to regional areas rather than a global review. The reason is that practice of what the authors call ‘FCNs” varies so much by region, it is hard to say that there is meaningful unified definition of what an FCN is globally, and this confuses the themes that emerge from the literature. One way to limit the scope of this review would be to focus the review only on FCNs in Europe, where perhaps there is a more clear definition and understanding of what an FCN is. This is somewhat addressed in the limitation section, where it is acknowledged that UK is overrepresented. Focusing specifically on Europe-based articles may strengthen this manuscript.
I have provided some examples from the manuscript where my major and minor concerns are, however, this is not an exhaustive list, as I found there to be too much instances to list. I highly recommend a revision of this literature review to revisit the scope, and also improve upon readability through editing of English language and more meticulous attention to formatting and detail (punctuation/grammar).
Major concerns:
- The manuscript purports to report on the state of the science of community and family nurse.
The search terms include “community nurse” and “family nurse”
The major issue is the search terms are not synonymous across countries/regions. The most predominant concern pertains to how the search term may lead to a mix of advanced nurses and generalists. The term “family nurse,” when used in the US context, returns articles pertaining to “family nurse practitioners.” These advanced practice nurses’ have vastly different skillsets than generalists nurses who merely practice nursing with families. Further, family practice nurses’ scope of practice differs by state in the US. These differences are not adequately captured or described in this article, and constitute a major flaw in its methods.
Example:
- 7 line 236-238. This is a specialized role of a family nurse practitioner, not a generalist nurse according to the citation’s title.
- It is not clear what a “family and community nurse” is. There is no definition provided. This is not a term that is regularly used in my country… the common term used is ‘community health nurse,’ and ‘public health nurse’ (and community/public health nurse, as these roles often overlap.) Looking through the literature reviewed, it does not seem that there is strong presence of articles pertaining to community health nurses. So, this would be a major limitation of this article in that it does not capture literature on CHN practice for which much is written.
Examples:
- 2 ln 76. How is a FCN defined and how does it differ from other roles of nursing? What is mentioned does not seem to differentiate FCN from any type of RN
- 2 ln 84. Authors must make a distinction between generalist RNs and Advance practice RNs (APRNs).
Minor concerns:
English language readability. There are many instances where it is not clear what is meant by the author due to the phrasing or word choice. Also there are some simple grammatical errors. I highly recommend the authors of the manuscript have the work reviewed by a native English speaker to assist in readability.
Examples:
- 1 line 40-45, run-on sentences, awkward phrasing.
- 1 45-46: what dues it mean ‘data can be great success’… this sentence is not clear what is beings tated.
- 2 ln 47: is the proper term elderly or ‘older adults?’ also, how do define this popoulatoin? >65 years old? >85 ?
- 2 ln49-51 awkward phrasing
p2 ln 54. What is a chronic patient? Patient with a chronic disease?
P2 ln 60-65 awkward phrasing.
Formatting. Headers do not seem to follow a conventional or predictable format.
Examples:
- 2 ln 76: why italicized?
- 7 line 219. Check header formatting style convention
Results/discussion mixed. There are some instances where discussion occurs in the results section. Results section should focus only on what was found. Discussion should focus on interpretation of the findings.
Examples:
- 7 line228-230 move to discussion
Other:
- 5 ln 204-205: What does in mean by “American” – USA, North America, South America? What does it mean by Asian continent? Does this include Japan? Why are some locations broken into countries (European – UK and Italy) while others refer to continents?
The article provides some interesting and relevant discussion points, specifically the discussion re Nursing Minimum Data Set (NMDS)
Author Response

(The authors gave the same response as above.)

Round 2
Reviewer 2 Report
The revised content might be good as it was very difficult to read. The authors should NOT provide a revised version that is readable.
Author Response
Dear reviewer,
Thank you for your suggestion. We provided a second round of proofreading.

Reviewer 3 Report
Hello, my initial comments have been appropriately addressed.
The English language issues have been significantly reduced. One term that I came across a few times is "territorial." I am unsure of the way the authors use this term (ln26, ln455, ln461) as it is not clear to me what it means in this context. I believe may signify non-hospital based care,community-based care, or 'public health'? For clarity, I recommend rewording the instances that this term is used.
The authors have addressed my main concern regarding how different regions use the term family/community nurse. I appreciate the citation for FCNs, and the discussion re the differences between community/family nurses in different regions, and the mention that further studies should focus within regions.
Author Response
Dear reviewer,
Thank you for this positive comment, and for your previous comments that were able to improve our manuscript.
Please find below point-by-point answers to your specific comments.
Thank you for your suggestion. We provide an additional English proofreading.
